# Dexamethasone is associated with early deaths in light chain amyloidosis patients with severe cardiac involvement

**Mélanie Bézard**[1,2]*, **Silvia Oghina**[1,2], **Damien Vitiello**[3,4], **Mounira Kharoubi**[1,2], **Ekaterini Kordeli**[4], **Arnault Galat**[1,2], **Amira Zaroui**[1,2], **Soulef Guendouz**[1,2], **Floriane Gilles**[1,2], **Jason Shourick**[5], **David Hamon**[2], **Vincent Audard**[6], **Emmanuel Teiger**[1,2,7], **Elsa Poullot**[7,8], **Valérie Molinier-Frenkel**[7,9], **François Lemonnier**[10], **Onnik Agbulut**[4], **Fabien Le Bras**[10], **Thibaud Damy**[1,2,7]

1 The French Cardiac Amyloidosis Reference Center, GRC Amyloid Research Institute, Réseau amylose Mondor, and DHU A-TVB, Créteil, France, 2 Cardiology Department, Assistance Publique des Hôpitaux de Paris, University Hospital Henri Mondor, Créteil, France, 3 Sport and Health Sciences Institute (I3SP–EA3625), Paris University, Paris, France, 4 Sorbonne University, Institut de Biologie Paris-Seine (IBPS), CNRS UMR 8256, Inserm ERL U1164, Adaptation biologique et vieillissement, Paris, France, 5 EA 7379, EpiDermE, UPEC, Créteil, France, 6 Nephrology and Transplantation Department, Rare Disease Reference Center «Syndrome Néphrotique Idiopathique», Assistance Publique des Hôpitaux de Paris, University Hospital Henri Mondor, Créteil, France, 7 Université Paris Est Créteil (UPEC), Institut National de la Santé et de la Recherche Médicale (INSERM) U955, Institut Mondor de Recherche Biomédicale (IMRB), Créteil, France, 8 Anatomy and Cytopathology, Assistance Publique des Hôpitaux de Paris, University Hospital Henri Mondor, Créteil, France, 9 Immunology Laboratory, Créteil, France, 10 Hematopathology-Lymphoid Unit, Assistance Publique des Hôpitaux de Paris, University Hospital Henri Mondor, Créteil, France

* melanie.bezard@aphp.fr

## Abstract

### Background

Cardiac light chain amyloidosis (AL-CA) patients often die within three months of starting chemotherapy. Chemotherapy for non-immunoglobulin M gammopathy with AL-CA frequently includes bortezomib (Bor), cyclophosphamide (Cy), and dexamethasone (D). We previously reported that NT-ProBNP levels can double within 24h of dexamethasone administration, suggesting a deleterious impact on cardiac function. In this study, we evaluate the role of dexamethasone in early cardiovascular mortality during treatment.

### Methods and findings

We retrospectively assessed 100 *de novo* cardiac AL patients (62% male, mean age 68 years) treated at our institute between 2009 and 2018 following three chemotherapy regimens: CyBorDComb (all initiated on day 1; 34 patients), DCyBorSeq (D, day 1; Cy, day 8; Bor, day 15; 17 patients), and CyBorDSeq (Cy, day 1; Bor, day 8; D, day 15; 49 patients). The primary endpoint was cardiovascular mortality and cardiac transplantation at days 22 and 455. At day 22, mortality was 20.6% with CyBorDComb, 23.5% with DCyBorSeq, and 0% with CyBorDSeq ($p = 0.003$). At day 455, mortality was not significantly different between regimens ($p = 0.195$). Acute toxicity of dexamethasone was evaluated on myocardial function using a rat model of isolated perfused heart. Administration of dexamethasone induced a decrease in left ventricular myocardium contractility and relaxation ($p<0.05$),

**Data Availability Statement:** Data can be shared upon request from ARMDC (Association pour la recherche multi-disciplinaire en cardiologie), Hôpital Henri Mondor, Service de cardiologie, unité

insuffisance cardiaque et amylose, 51 avenue du maréchal de Lattre de Tassigny, 94000 Créteil, thibaud.damy@aphp.fr.

**Funding:** This work was supported by the ARMDC (Association pour la Recherche Multi-Disciplinaire en Cardiologie), a non-profit organization. The funders had no role in study design, data collection and analysis, decision to publish, or preparation of the manuscript.

**Competing interests:** Pr Vincent Audard received consulting fees from Addmedica not related to the submitted work. Dr Silvia Oghina reported personal fees from Pfizer, outside of the submitted work. Pr Thibaud Damy received grant and/or consulting fees from PFIZER, AKCEA, ALNYLAM, PROTHENA, and JANSSEN outside the submitted work. The other authors declared no conflict of interests. This does not alter our adherence to PLOS ONE policies on sharing data and materials.

**Abbreviations:** AL, Light chain amyloidosis; CA, cardiac amyloidosis; CyBorDComb, Cyclophosphamide Bortezomib Dexamethasone Combined; CyBorDSeq, Cyclophosphamide Bortezomib Dexamethasone administered Sequentially; DCyBorSeq, Dexamethasone Cyclophosphamide Bortezomib administered Sequentially; FLC, free light chain; NT-proBNP, N-terminal pro-brain natriuretic peptide; OS, Overall survival; TnT-HS, Troponin T High sensitivity.

supporting a potential negative inotropic effect of dexamethasone in AL-CA patients with severe cardiac involvement.

## Conclusion

Delaying dexamethasone during the first chemotherapy cycle reduces the number of early deaths without extending survival. It is clear that dexamethasone is beneficial in the long-term treatment of patients with AL-CA. However, the initial introduction of dexamethasone during treatment is critical, but may be associated with early cardiac deaths in severe CA. Thus, it is important to consider the dosage and timing of dexamethasone introduction on a patient-severity basis. The impact of dexamethasone in the treatment of AL-CA needs further investigation.

## Introduction

Amyloidosis is characterized by the abnormal accumulation of amyloid fibrils in organs and tissues [1, 2], due to protein misfolding. In light chain amyloidosis (AL), the clonal proliferation of plasma cells in the bone marrow is responsible for excessive production of free immunoglobulin light chains (FLC) with amyloidogenic properties [1, 2]. Amyloid FLC fibrils deposit in tissues, most frequently in the heart, kidney, and peripheral nervous system [1, 3, 4], where they cause organ dysfunction. In the heart, amyloid deposits thicken the cardiac walls and diminish cardiac function [1, 5].

If cardiac involvement is diagnosed in AL patients, mortality is rapid, with a median survival of 6 months, if untreated [1, 2]. In 2004, the Mayo Clinic reported a staging system to classify the severity of cardiac damage using two cardiac biomarkers, serum cardiac troponin-T and N-terminal pro-brain natriuretic peptide (NT-proBNP) [6]. In 2012, the Mayo Clinic staging was updated [7]. More recently another European staging was proposed and established the stages IIIa and IIIb depending on the level of NT-proBNP [8, 9].

The use of cyclophosphamide (Cy), bortezomib (Bor), and dexamethasone (D), the CyBorD regimen, to treat AL amyloidosis, caused by a non-immunoglobulin M (IgM) monoclonal gammopathy, has significantly improved survival outcomes [10]: CyBorD results in a complete hematologic response of 71% and a 1-year OS of 65% [11]. However, serious and often fatal cardiac events can occur during the first cycle of treatment, including sudden death, syncope, arrhythmia, and heart failure [12].

We recently reported a pilot study, showing that AL cardiac amyloidosis patients treated with sequential DCyBor regimen had significantly increased levels of the heart failure biomarker NT-proBNP after each dexamethasone administration [13]. The NT-proBNP increased levels were unrelated to FLC release or cardiac ischemia, since FLC and troponin-T levels were unchanged.

These observations led us to investigate dexamethasone's role in early cardiovascular mortality of AL cardiac amyloidosis patients. In the present study, patients were treated with several CyBorD regimens differing in the timing of dexamethasone administration.

## Methods

### Study population

All patients gave informed consent for the use of their anonymized data. The study complied with the Declaration of Helsinki and was approved by the French data protection authority (Comité National de l'Informatique et des Libertés, CNIL, #1431858).

We enrolled all patients with confirmed cardiac AL, referred to the GRC Amyloid Research Institute in Creteil, France, between 2009 and 2018. Their medical records were retrospectively reviewed. Eligible patients had newly diagnosed cardiac AL with an associated non-IgM monoclonal gammopathy, when CyBorD was initiated. Patients previously treated with chemotherapy or with chemotherapy other than CyBorD, as well as those in cardiogenic shock or in palliative care were ineligible.

## Diagnostic criteria for AL cardiac amyloidosis

All patients had a medical evaluation when they arrived at the institute, including clinical and laboratory tests (complete blood count, serum NT-proBNP, and high-sensitivity troponin-T), electro- and echocardiography, [99m]-Tc-HMDP scintigraphy, and cardiac magnetic resonance imaging.

AL was diagnosed by histological analysis of a tissue biopsy (extracardiac or endomyocardial) showing amyloid deposits specifically Congo red stained and labelled by an antibody directed to kappa or lambda FLC, and negative with antibody directed to TTR. Monoclonal gammopathy was diagnosed by electrophoresis and immunofixation of serum and urine, and quantification of circulating kappa and lambda FLC. Patients included had either kappa or lambda FLC levels above reference values, associated with the kappa/lambda ratio indicative of a monoclonal FLC. Multiple myeloma was defined in accordance with the International Myeloma Working Group guidelines [14]. Creatinine was assessed to estimate the glomerular filtration rate according to Chronic Kidney Disease Epidemiology Collaboration (CKD-EPI) equation. Cardiac involvement was confirmed by transthoracic ultrasonography with interventricular septal hypertrophy ($>12$ mm) and increased levels of NT-proBNP above 332 ng/L (in the absence of renal failure or atrial fibrillation). Cardiac amyloidosis severity was assessed using cardiac biomarkers: NT-proBNP and high-sensitivity troponin-T (TnT-HS). Patients were classified by European version of Mayo Clinic stages referred as European stages: stage I, no cardiac biomarker elevated; stage II, one biomarker elevated; stage IIIa, both biomarkers elevated with NT-proBNP $>332$ ng/L and TnT-HS $>50$ ng/L; and stage IIIb, NT-proBNP $>8500$ ng/L and TnT-HS $>50$ ng/L.

## Therapeutic regimens

Between 2009 and 2018, three CyBorD regimens successively used at our institute varied with respect to the dosing and timing of dexamethasone. Between 2009 and 2015, the CyBorD-Comb regimen was in common use. The regimen concurrently administered cyclophosphamide (on days 1, 8, and 15), bortezomib (on days 1, 8, 15, and 22), and dexamethasone (on days 1, 2, 8, 9, 15, 16, 22, and 23). The CyBorDComb regimen was gradually replaced, between 2013 and 2015, by the DCyBorSeq regimen. The DCyBorSeq regimen sequentially administered dexamethasone (on days 1, 2, 8, 9, 15, 16, 22, and 23), then cyclophosphamide (on days 8 and 15) and bortezomib (on days 15 and 22). In 2016, CyBorDSeq became the standard treatment at our institute [13]. CyBorDSeq sequentially administered cyclophosphamide (on days 1, 8, and 15), bortezomib (on days 8, 15, and 22), with a delayed administration of dexamethasone (on days 15, 22, and 23). In the CyBorD regimens, cyclophosphamide (300 mg/m$^2$) was administered orally, bortezomib (1.3 mg/m$^2$) subcutaneously, and dexamethasone orally at a dose of either 10, 20, or 40 mg, adapted to cardiac severity.

## Statistical analysis

Continuous variables were expressed as median with the interquartile range (IQR) and dichotomous data as percentages. Frequencies for quantitative variables were compared using the

Chi$^2$ test with Pearson's correction. For continuous data, two groups were compared using the Mann-Whitney test and more than two groups the Kruskal-Wallis test.

Follow-up data were obtained from medical files. Survival was calculated from the date of treatment initiation until the date of death, cardiac transplantation, or last follow-up.

For the survival analysis, the 16 patients alive without follow-up of at least 455 days at the time of analysis were not assessed in univariate and multivariate survival analysis but were included in the Cox model and censored at the time of last follow-up.

To access the impact of the timing of dexamethasone administration, the data of patients who received CyBorDComb and DCyBorSeq were pooled (early administration) and compare to patients who received CyBorDSeq (late administration).

Patients were analyzed for mortality and censored at the date of last follow-up or death. Survival analysis (univariate) was performed by Kaplan-Meier method and compared by log-rank test. A Cox model was developed to assess the association between CyBorD regimen administered and mortality. A Cox model with time-varying effects was performed to analyze the effect of dexamethasone administration on mortality. The time varying effect involved coding patients as negative for dexamethasone before their first dexamethasone administration and positive thereafter. Proportionality of risk was assessed using Schoenfeld's residuals.

The statistical analyses were performed using the SPSS software (v19.0 for Windows 2010 SPSS Inc.) and R version 3.6.1. A p-value below 0.05 would be considered statistically significant.

## Isolated perfused heart preparation

Fourteen female Wistar rats (Janvier Labs, Genest Saint Isle, France), weighing 225–250 g, were used in this study. All procedures were performed according to National and European legislations, in conformity with the Public Health Service Policy on Human Care and Use of Laboratory Animals and were approved by our institutional Ethics Committee "Charles Darwin" (Sorbonne University). Rats were anesthetized under isoflurane (induced and maintained with 3–4%). Sodium heparin (500 IU) was administered in the femoral vein. The heart was quickly removed and placed under constant pressure (60 mmHg), in the Langendorff perfused heart system (Harvard Apparatus, Les Ulis, France). The ascending aorta was cannulated and retrogradely perfused with Krebs buffer (118 mM NaCl, 5 mM KCl, 0.9 mM MgSO$_4$, 1.2 mM KH$_2$PO$_4$, 25 mM NaHCO$_3$, 11 mM d-glucose, and 2.5 mM CaCl$_2$), equilibrated with 95% O$_2$-5% CO$_2$ at 37˚C. A water-filled balloon tied to a microtip catheter pressure transducer was introduced into the left ventricular cavity to normalize the diastolic blood pressure and to record cardiac function. The hearts were not electrically paced. During a 15 to 20-min stabilization period, left ventricular (LV) end-systolic pressure (LVP$_{max}$), LV end-diastolic pressure (LVP$_{min}$), LV developed pressure (LVDP: index of global LV function), LV minimal rate of pressure decline (dP/dt$_{min}$, index of LV relaxation), LV maximal rate of pressure development (dP/dt$_{max}$, index of LV contractility) and heart rate were automatically and continuously recorded by the ISOHEART software (Harvard Apparatus, Les Ulis, France). After basal levels were recorded, data were collected over 15–20 min perfusion of 10 μM dexamethasone sodium phosphate (DEXAMETHASONE Mylan 4 mg/mL) at 0.3 mL/min.

Groups were statistically compared with GraphPad Prism 8 (GraphPad Software Inc., SD, USA) using paired t-test. If normal distribution (verified by Shapiro-Wilk's test) was not assumed, the groups were compared using the Wilcoxon test. Values are given as means with associated standard error of means.

## Results

### Patient and cardiac amyloidosis characteristics

Overall, 273 cardiac amyloidosis patients were referred to our institute between 2009 and 2018 for AL amyloidosis, and 100 patients were included (Fig 1). The characteristics of the patients are shown in Table 1. Patients were predominantly males (62%) with a median age at diagnosis of 66 years (IQR: 59–73). Most patients, 77 (77%), had AL amyloidosis of lambda subtype. The NT-proBNP and TnT-HS levels, and consequently the severity of cardiac amyloidosis were not significantly different between CyBorD regimens (Table 1). The first dose of dexamethasone received during the first cycle of chemotherapy were significantly different between the different treatment groups (p<0.001; Table 1). Due to our experience, we questioned the toxicity of dexamethasone, and so we adapted and decreased the first doses of dexamethasone administered in time, that is why the first dose of dexamethasone received during the first cycle of chemotherapy were significantly different between the different treatment groups.

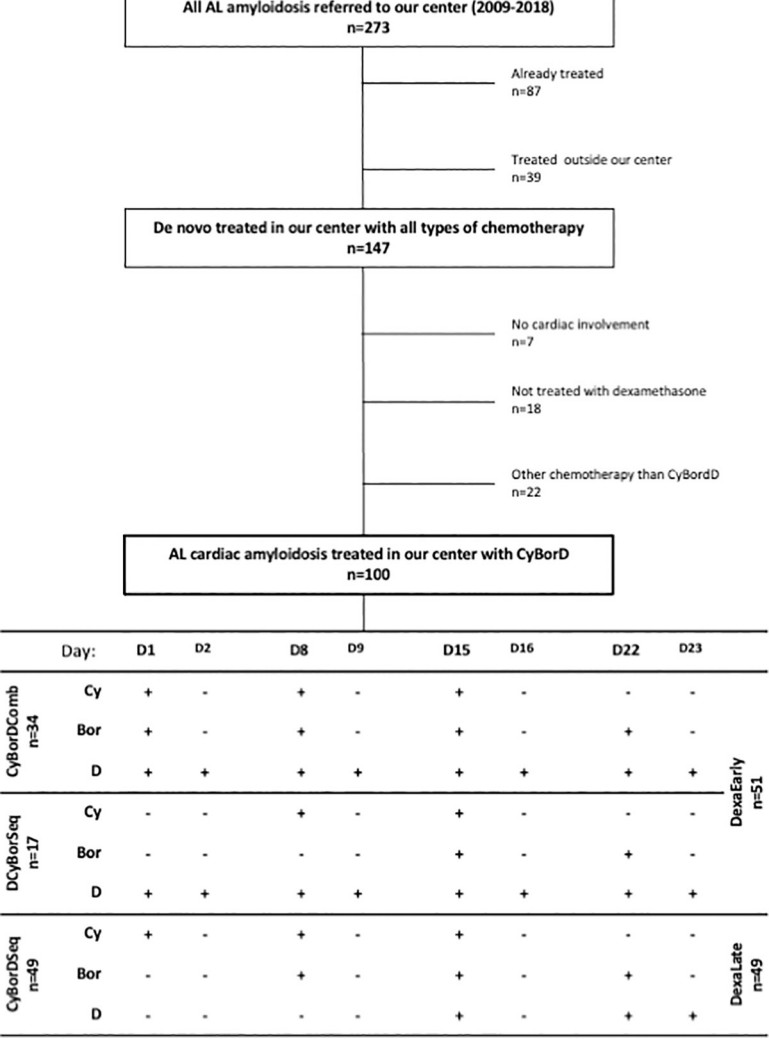

**Fig 1. Flow chart.** The figure describes the selection of studied population among AL patients and the different groups of treatment based on the time of administration of chemotherapy.

**Table 1. Baseline characteristics in treated patients.**

| Variables | Overall | CyBorDComb | DCyBorSeq | CyBorDSeq | p-value |
|---|---|---|---|---|---|
| | n = 100 | n = 34 | n = 17 | n = 49 | |
| **Dexamethasone first dose, mg** | 20 (20; 40) | 20 (20; 40) | 20 (20; 40) | 20 (20; 20) | **<0.001** |
| **Clinical characteristics** | | | | | |
| Age, years | 66 (58; 73) | 68 (62; 73) | 64 (56; 68) | 64 (54; 74) | 0.294 |
| Male, n (%) | 62 (62) | 20 (59) | 9 (53) | 33 (67) | 0.514 |
| BMI, kg/m$^2$ | 23.8 (21.8; 26.4) | 23.7 (21.7; 25.2) | 22.6 (20.5; 24.5) | 24.5 (22.15; 27.1) | 0.239 |
| **CV characteristics** | | | | | |
| NYHA class III-IV vs I-II, n (%) | 48 (51) | 16 (50) | 9 (56) | 23 (49) | 0.878 |
| Heart rate, beats/min | 79 (70; 88) | 76 (69; 85) | 83 (78; 94) | 80 (70; 88) | 0.121 |
| Systolic blood pressure, mmHg | 98 (88; 111) | 102 (93; 123) | 96 (87; 103) | 98 (88; 111) | 0.122 |
| Diastolic blood pressure, mmHg | 63 (56; 70) | 65 (59; 73) | 59 (55; 67) | 62 (55; 70) | 0.118 |
| Atrial fibrillation, n (%) | 16 (16) | 5 (15) | 5 (29) | 6 (12) | 0.309 |
| ICD, n (%) | 49 (49) | 11 (32) | 10 (59) | 28 (57) | 0.057 |
| **CV Risk factors** | | | | | |
| Diabetes, n (%) | 17 (17) | 8 (24) | 2 (12) | 7 (14) | 0.446 |
| Dyslipidemia, n (%) | 29 (29) | 16 (47) | 5 (29) | 8 (16) | **0.010** |
| Hypertension, n (%) | 36 (36) | 18 (53) | 1 (6) | 17 (35) | **0.004** |
| **Biology variables** | | | | | |
| NT-proBNP, pg/mL | 5105 (2592; 11428) | 5638 (2367; 12829) | 4000 (3325; 10534) | 5501 (2332; 9850) | 0.938 |
| Troponin T HS, ng/mL | 98.0 (65.5; 143.5) | 98.0 (64.0; 172.0) | 90.5 (43.8; 147.3) | 101.0 (70.0; 142.5) | 0.778 |
| Hemoglobin, g/dL | 12.8 (11.1; 13.7) | 12.8 (10.3; 13.5) | 12,9 (11.9; 13.3) | 12.7 (11.2; 13.8) | 0.580 |
| Calcemia, mmol/L | 2.31 (2.21; 2.43) | 2.28 (2.07; 2.34) | 2.35 (2.26; 2.55) | 2.35 (2.21; 2.44) | **0.015** |
| Creatinine, µmol/L | 94.0 (79.3; 130.5) | 113.5 (85.0; 184.5) | 85.0 (70.0; 105.5) | 93.0 (74.0) | **0.019** |
| eGFR, mL/min/1.73 m$^2$ | 58 (44; 83) | 52 (30; 68) | 59 (44; 87) | 66 (51; 90) | **0.025** |
| Bilirubin, µmol/L | 6.0 (4.0; 10.0) | 8.0 (4.8; 13.0) | 7.0 (4.0; 13.5) | 5.0 (4.0; 8.0) | **0.026** |
| **Amyloid characteristics** | | | | | |
| European stage | | | | | 0.232 |
| II, n (%) | 11 (11) | 6 (18) | 1 (6) | 4 (8) | |
| IIIa, n (%) | 56 (56) | 14 (41) | 12 (71) | 30 (61) | |
| IIIb, n (%) | 33 (33) | 14 (41) | 4 (24) | 15 (31) | |
| Medullar plasmocytes, % | 8 (6; 15) | 7 (5; 17) | 7 (6; 13) | 8 (6; 15) | 0.667 |
| Lambda, n (%) | 77 (77) | 25 (74) | 12 (71) | 40 (82) | 0.216 |
| dFLC, mg/L | 259 (122; 484) | 299 (118; 708) | 343 (215; 529) | 205 (115; 455) | 0.247 |
| **CRAB criteria** | | | | | |
| Anemia, n (%) | 36 (36) | 14 (41) | 3 (18) | 19 (39) | 0.218 |
| Kidney disease, n (%) | 20 (20) | 12 (35) | 2 (12) | 6 (12) | **0.023** |
| Hypercalcemia, n (%) | 2 (2) | 0 (0) | 2 (17) | 0 (0) | **0.001** |
| Bone lesions, n (%) | 6 (6) | 1 (3) | 0 (0) | 5 (10) | **<0.001** |
| **Echocardiography characteristics** | | | | | |
| LVEF, % | 50 (42; 59) | 49 (40; 60) | 44 (35; 48) | 52 (45; 60) | **0.010** |
| IVST, mm | 15.0 (13.0; 17.0) | 14.0 (13.0; 16.3) | 14.0 (12.3; 17.8) | 16.0 (14.0; 17.0) | 0.130 |
| GL Strain, % | 9.4 (7.8; 12.0) | 9.6 (7.4; 13.9) | 7.9 (6.7; 9.1) | 10.0 (8.3; 11.9) | 0.072 |
| LVEDD, mm | 42.0 (38.0; 47.0) | 43.5 (39.8; 48.3) | 40.0 (36.3; 45.8) | 42.0 (38.0; 46.5) | 0.171 |
| LVESD, mm | 29.0 (25.0; 33.0) | 29.5 (24.5; 33.8) | 28.0 (22.0; 34.0) | 29.0 (26.0; 32.0) | 0.856 |
| LV mass, g | 245 (210; 307) | 241 (209; 289) | 217 (187; 364) | 261 (223; 317) | 0.162 |

(*Continued*)

**Table 1.** (Continued)

| Variables | Overall | CyBorDComb | DCyBorSeq | CyBorDSeq | p-value |
|---|---|---|---|---|---|
| | n = 100 | n = 34 | n = 17 | n = 49 | |
| Deceleration time, ms | 140 (105; 182) | 150 (119; 217) | 108 (87; 164) | 142 (106; 182) | 0.062 |

Values are median (interquartile range)

CyBorD = Cyclophosphamide, bortezomib, and dexamethasone; BMI = body mass index; dFLC = free light-chain difference (dFLC = |kappa-lambda|);

eGFR = estimated glomerular filtration rate; GL = global longitudinal; ICD = implantable cardioverter-defibrillator; IVST = interventricular septum thickness;

LVEDD = left ventricular end-diastolic diameter; LVEF = left ventricular ejection fraction; LVESD = left ventricular end-systolic diameter; NT-proBNP = N-terminal

pro-B-type natriuretic peptide; NYHA = New York Heart Association.

However, the first dose of dexamethasone was similar between the group of living patients and the group of patients who have died (p = 0.419; Table 2).

## Survival according treatment initiation year

Between 2009 and 2012, 11 patients were treated with a CyBorD regimen: 10 with CyBorD-Comb and 1 with DCyBorSeq (S1 Fig). Similarly, between 2013 and 2015, 36 patients were treated: 18 with CyBorDComb, 16 with DCyBorSeq, and 2 with CyBorDSeq, and finally, between 2016 and 2018, 53 patients were treated: 6 with CyBorDComb, and 47 with CyBorD-Seq. The median survival of patients initiating treatment during the intervals were 317 days between 2009–2012, and not reached between both 2013–2015 and 2016–2018 (S1 Fig). The year of cardiac AL amyloidosis diagnosis was not significantly associated with survival, p = 0.576.

## Survival according to severity of cardiac amyloidosis

Concerning severity of cardiac amyloidosis, 11 patients (11%) were European stage II, 56 (56%) were IIIa, and 33 (33%) stage IIIb (Table 1). Median survival was significantly shorter in patients with European stage IIIb: 201 days (95% CI: 16.86–385.14), compared to those with stages II or IIIa (pooled data): median not reached, p = 0.002 (S1 Fig).

## Survival according to the timing of dexamethasone administration, early or late in the cycle

During the first 28-day cycle of chemotherapy, survival was significantly prolonged in patients when dexamethasone was administered late in the cycle on days 15, 22, and 23, p = 0.003 (Fig 2A). Indeed, no deaths were reported during cycle-1 of CyBorDSeq, in contrast to the 7 deaths reported during cycle-1 of CyBorDComb and the 4 deaths during cycle-1 of DCyBorSeq. However, survival was not significantly different between CyBorD regimens during the second and third treatment cycles, p = 0.116 (Fig 2B).

All early deaths during the cycle-1 of chemotherapy were cardiovascular-related (Fig 3A–3C). During cycle-1 of CyBorDComb, 5 sudden deaths occurred: one patient on days 1, 3, and 4 and two patients on day 10 (Fig 3A). In addition, 2 patients died of worsening heart failure on day 16 and 17. Similarly, during cycle-1 of DCyBorSeq, 4 early deaths due to worsening heart failure (Fig 3B). In contrast, during cycle-1 of CyBorDSeq, no deaths occurred. However, a sudden death occurred on day 29, 15 days after initiating and 7 days after the last dose of dexamethasone (Fig 3C). All patients who died during the first cycle of chemotherapy were in stage III (a or b) of the European score, the cumulative doses of dexamethasone are very variable depending on the patient (S1 Table).

**Table 2. Characteristics of living and deceased patient after 455 days of complete follow-up.**

| Variables | Overall | Alive | Dead | p-value |
|---|---|---|---|---|
| | n = 84 | n = 41 | n = 43 | |
| Dexamethasone first dose, mg | 20 (20; 40) | 20 (20; 40) | 20 (20; 40) | 0.419 |
| **Clinical characteristics** | | | | |
| Age, years | 66 (59; 74) | 70 (61; 75) | 64 (55; 68) | **0.018** |
| Male, n (%) | 53 (63) | 27 (66) | 26 (60) | 0.609 |
| BMI, kg/m$^2$ | 23.8 (22.0; 26.4) | 24.0 (22.2; 25.7) | 23.7 (21.3; 26.6) | 0.760 |
| **CV characteristics** | | | | |
| NYHA class III-IV vs I-II, n (%) | 43 (54) | 18 (46) | 25 (63) | 0.145 |
| Heart rate, beats/min | 79 (70; 88) | 79 (72; 88) | 79 (69; 88) | 0.680 |
| Systolic blood pressure, mmHg | 96 (88; 106) | 100 (88; 108) | 93 (87; 102) | 0.130 |
| Diastolic blood pressure, mmHg | 62 (56; 67) | 62 (54; 68) | 63 (58; 67) | 0.517 |
| Atrial fibrillation, n (%) | 15 (18) | 6 (15) | 9 (21) | 0.460 |
| ICD, n (%) | 41 (49) | 24 (59) | 17 (40) | 0.082 |
| **CV Risk factors** | | | | |
| Diabetes, n (%) | 14 (17) | 9 (22) | 5 (12) | 0.204 |
| Dyslipidemia, n (%) | 27 (32) | 17 (41) | 10 (23) | 0.074 |
| Hypertension, n (%) | 31 (37) | 17 (41) | 14 (33) | 0.398 |
| **Biology variables** | | | | |
| NT-proBNP, pg/mL | 5502 (3295; 12408) | 3952 (2607; 7796) | 6938 (4084; 16463) | **0.003** |
| Troponin T HS, ng/mL | 103 (64; 161) | 79 (55; 134) | 126 (89; 198) | **0.005** |
| Hemoglobin, g/dL | 12.7 (11.1; 13.5) | 12.6 (11.0; 13.3) | 12.9 (11.2; 13.8) | 0.233 |
| Calcemia, mmol/L | 2.30 (2.17; 2.42) | 2.26 (2.10; 2.39) | 2.32 (2.21; 2.44) | 0.107 |
| Creatinine, µmol/L | 98 (80; 137) | 88 (77; 130) | 106 (89; 143) | 0.151 |
| eGFR, mL/min/1.73m$^2$ | 57.4 (42.6; 78.5) | 53.8 (37.5; 90.9) | 58.1 (44.4; 74.5) | 0.936 |
| Bilirubin, µmol/L | 6.0 (4.0; 10.8) | 5.0 (4.0; 8.5) | 8.0 (4.0; 13.0) | **0.017** |
| **Amyloid characteristics** | | | | |
| European stage | | | | 0.107 |
| II, n (%) | 7 (8) | 4 (10) | 3 (7) | |
| IIIa, n (%) | 47 (56) | 27 (66) | 20 (47) | |
| IIIb, n (%) | 30 (36) | 10 (24) | 20 (47) | |
| Medullar plasmocytes, % | 8 (6; 14) | 8 (6; 14) | 10 (6; 16) | 0.711 |
| Lambda, n (%) | 63 (75) | 33 (80) | 30 (70) | 0.387 |
| dFLC, mg/L | 278 (122; 509) | 259 (123; 681) | 332 (120; 452) | 0.652 |
| **CRAB criteria** | | | | |
| Anemia, n (%) | 32 (38) | 18 (44) | 14 (33) | 0.285 |
| Kidney disease, n (%) | 19 (23) | 10 (24) | 9 (21) | 0.705 |
| Hypercalcemia, n (%) | 2 (3) | 2 (6) | 0 (0) | 0.146 |
| Bone lesions, n (%) | 5 (6) | 2 (5) | 3 (7) | 0.713 |
| **Echocardiography characteristics** | | | | |
| LVEF, % | 49 (41; 59) | 51 (45; 60) | 45 (38; 57) | **0.042** |
| IVST, mm | 15 (13; 17) | 14 (13; 16) | 16 (14; 18) | **0.005** |
| GL Strain, % | 9.0 (7.4; 11.7) | 10.4 (8.2; 14.1) | 8.6 (6.5; 9.8) | **0.003** |
| LVEDD, mm | 42 (38; 47) | 45 (39; 48) | 40 (38; 45) | **0.043** |
| LVESD, mm | 29 (25; 33) | 29 (24; 33) | 29 (25; 33) | 0.821 |
| LVmass, g | 242 (206; 303) | 237 (199; 294) | 261.1 (219.5; 307.0) | 0.149 |

*(Continued)*

**Table 2.** (Continued)

| Variables | Overall | Alive | Dead | p-value |
|---|---|---|---|---|
| | **n = 84** | **n = 41** | **n = 43** | |
| Deceleration time, ms | 137 (104; 187) | 133 (95; 181) | 144 (108; 193) | 0.549 |

Values are median [(interquartile range)

CyBorD = Cyclophosphamide, bortezomib, and dexamethasone; BMI = body mass index; dFLC = free light-chain difference (dFLC = |kappa-lambda|);
eGFR = estimated glomerular filtration rate; GL = global longitudinal; ICD = implantable cardioverter-defibrillator; IVST = interventricular septum thickness;
LVEDD = left ventricular end-diastolic diameter; LVEF = left ventricular ejection fraction; LVESD = left ventricular end-systolic diameter; NT-proBNP = N-terminal
pro-B-type natriuretic peptide; NYHA = New York Heart Association.

## The predictive value of patient and cardiac amyloidosis characteristics for survival

Characteristics of the patients who died or alive during the 455 days (15 months) after initiating treatment are shown in Table 2. Death seems to be associated with younger age, which

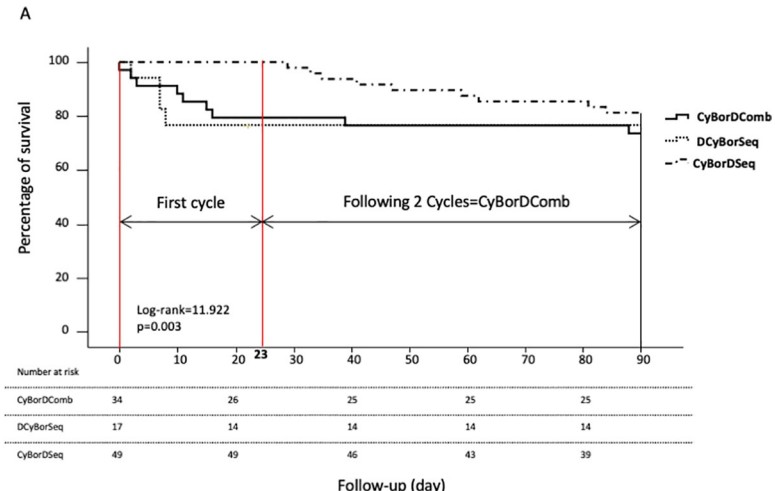

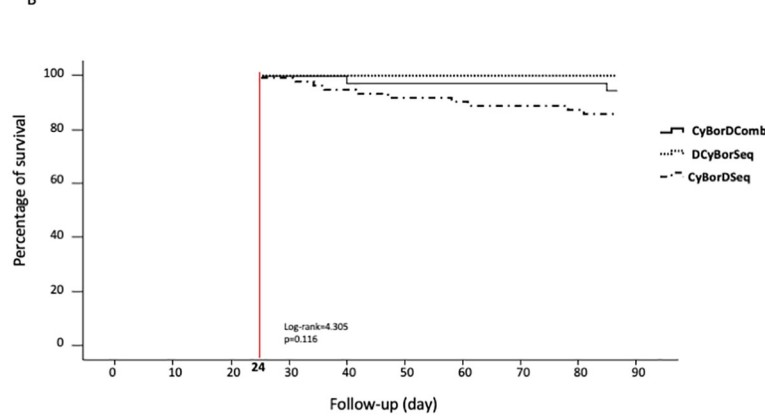

**Fig 2. Survival curves of patients according to the sequence of the chemotherapy's administration.** (A) Survival of patient within 90 days of initiating chemotherapy. (B) Survival of patient within the 2nd and 3rd cycles of chemotherapy depending on the chemotherapy regimen administered.

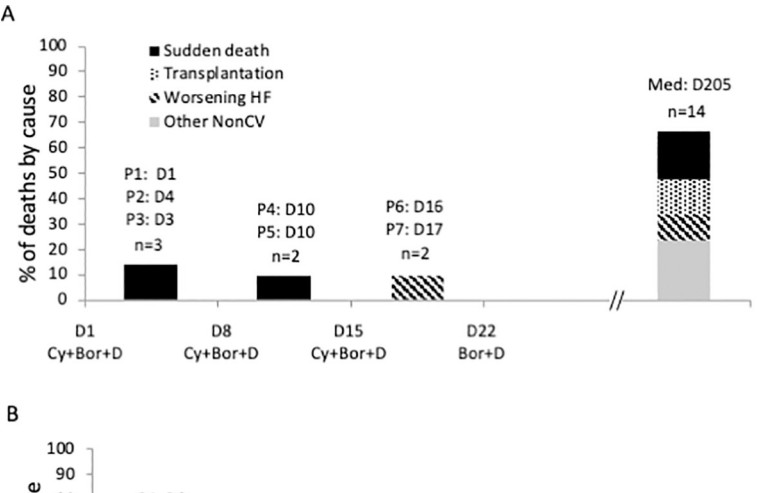

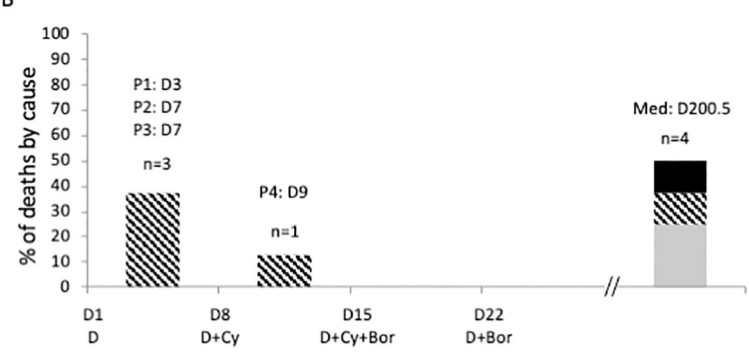

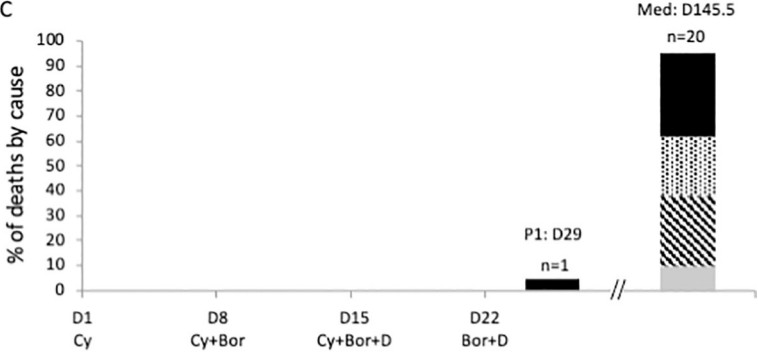

**Fig 3. Prevalence of causes of deaths after each chemotherapy administration.** Prevalence of causes of cardiovascular deaths with the CyBorDComb (A), DCyBorSeq (B), and CyBorDSeq (C) regimens.

may be due to younger patients suffer from diagnostic delay and more severe and aggressive amyloidosis. Death was significantly associated with higher NT-proBNP (6938 pg/mL versus 3952 pg/mL; p = 0.003) and TnT-Hs levels (126 ng/mL versus 79 ng/mL; p = 0.005), and global longitudinal strain by echocardiography (-8.6% versus -10.4%; p = 0.003) (Table 2). In the multivariate Cox regression analysis, time of dexamethasone administration was significantly associate with an increased risk of mortality (HR = 43.04; 95% CI: 5.93–312.58; p<0.001), Fig 4.

### Isolated perfused heart results

Using the isolated and perfused heart model, we investigated cardiac function under basal and dexamethasone perfusion conditions. During dexamethasone administration, the $LVP_{min}$ was

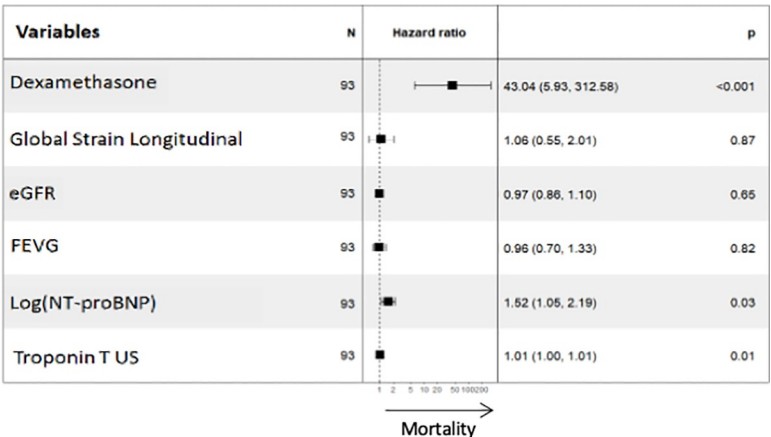

**Fig 4. Multivariate Cox regression analysis.** Forest plot of the multivariate Cox regression analysis of the risk factors associated with mortality.

significantly increased from 10 to 16 mmHg ($p > 0.01$), while $LVP_{max}$ remained unchanged. LVPD was therefore significantly decreased showing a global dysfunction of the left ventricle (LV) (Fig 5A). Accordingly, both LV contractility (dP/dt$_{max}$: 812 versus 635 mmHg/s;

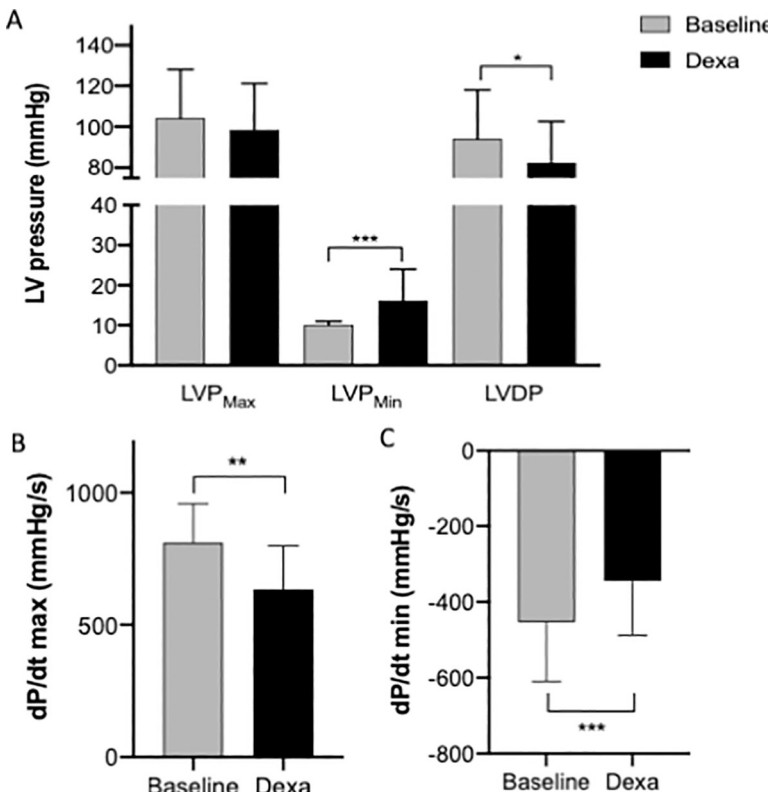

**Fig 5. Acute effect of dexamethasone perfusion on left ventricular global function and myocardial function in a constant pressure Langendorff isolated perfused rat heart model.** Left ventricular developed pressure (LVDP), LV end-systolic pressure (LVPmax), LV end-diastolic pressure (LVPmin), dP/dtmax and dP/dtmin were automatically and continuously recorded in an isolated and perfused rat heart before and after dexamethasone perfusion (10 μM; 0.3 L.min-1). Data are presented as means ± SEM. *: $p < 0.05$, **: $p < 0.01$, ***: $p < 0.001$. n = 14.

p<0.001) and LV relaxation (dP/dt$_{min}$: -453 versus -345 mmHg/s; p<0.01) appeared significantly decreased when dexamethasone was administered to healthy heart (Fig 5B and 5C).

## Discussion

In AL-CA patients, poor prognosis correlates with cardiac severity, with deaths frequently occurring during the first months of chemotherapy. Currently, the most frequently used chemotherapy for non-IgM AL-CA combines bortezomib, dexamethasone, and cyclophosphamide, and has in general improved survival in patients with cardiac amyloidosis [10]. However, patients with severe CA were often not eligible for trials assessing this chemotherapy. Consequently, data to analyze the effect of this combination in patients with severe cardiac involvement is lacking. Our retrospective study explored the use and safety of dexamethasone in patients with European severe AL-CA. We confirmed the poor prognosis of these patients and show that dexamethasone is associated with early cardiac-related deaths.

In 2013, in the European study, 346 patients were included of whom 338 (97%) had a cardiac involvement [8]. The OS at one month was about 90%. Furthermore, a prospective observational study enrolled 915 patients with newly diagnosed AL treated with a bortezomib-based chemotherapy; of these, 653 patients (71%) had cardiac involvement [15]. The study reported a median OS of 4 months for patients Mayo Clinic (2012) stage IV. The OS at one month was about 80%. Similarly, an international retrospective study analyzed 60 AL patients with cardiac involvement (Mayo Clinic (2004) stage III) [10]. However, after a median follow-up of 11.8 months the median OS had not been reached. Our median OS results are comparable with those reported in these studies suggesting that our cohort is representative of CA patients.

We needed to establish whether the administered chemotherapy regimen impacts the incidence of early deaths in severe CA patients. However, in cardiac amyloidosis clinical trials all patients receive a treatment: the use of a placebo is unethical. Finally, since cardiac amyloidosis is a severe disease, trials assessing chemotherapy tend to enroll patients with mild cardiac amyloidosis, with the results extrapolated to patients with severe disease. Consequently, the safety and efficacy of dexamethasone alone and in combination has been poorly studied in patients with severe CA. A causal relationship between dexamethasone and death was difficult to establish since dexamethasone was always combined with other drugs such as cyclophosphamide and bortezomib. Here we show that significantly more early deaths occurred when dexamethasone, combined or alone, was administered earlier (the CyBorDComb and DCyBorSeq regimens) compared to later in the first cycle (the CyBorDSeq regimen). Furthermore, in our study all early deaths occurring in the first 28-day cycle of chemotherapy were cardiovascular-related and most occurred 3–4 days after dexamethasone administration. However, among the patients who did not received dexamethasone, 3 patients received only cyclophosphamide, 2 of them died before day 15 (cardiorespiratory arrest), 8 patients received velcade and cyclophosphamide, 2 of them died before day 15 (infection and cardiogenic shock).

### Mode of death in severe cardiac amyloidosis following chemotherapy administration

Severe and often fatal cardiac events, including arrhythmias, sudden deaths, syncope, and heart failure, frequently occur during the first cycle of CyBorD regimens [13]. Indeed, a study assessed cardiac arrhythmias in patients with severe cardiac AL. Within 24h of baseline assessments, loop recorders were implanted in 20 patients: 7 were Mayo Clinic stage III and 13 were stage IV [16]. The median OS, from date of implant, was 61 days. During the study 13 patients died: 3 (23%) stage III and 10 (77%) stage IV. In patients that died, the weekly recordings during treatment failed to identify fatal dysrhythmias except for those recorded immediately

before cardiac arrest. In 8/13 patients the recordings prior to cardiac arrest were retrieved and all the initial changes in cardiac rhythm were bradycardia ($\leq$35 bpm). All 20 patients received a dexamethasone-based chemotherapy: 18 the CyBorD regimen, one combined with lenalidomide, and one with melphalan. Interestingly, most patients died between 48 h to 6 days after the second cycle of chemotherapy and died of sinus bradycardia followed by complete atrioventricular block, atrial fibrillation or flutter. These observations led the authors to suspect that the chemotherapy administered to treat systemic AL worsened prognosis in patients with severe cardiac infiltration either by promoting arrhythmias or affecting systolic function.

## Cardiac toxicity of chemotherapy in severe cardiac amyloidosis

Overall, haematological treatment of AL-CA aims to prevent the buildup of amyloid deposits while simultaneously preventing congestive heart failure [17]. However, dexamethasone is known to induce cardiotoxicity [18]. Several cardiovascular disorders result from extended use of dexamethasone, including cardiac fibrosis, an increase in noradrenaline-induced vascular contraction, increased apoptosis, and decreased angiogenesis [18, 19]. We previously investigated the cardiac effects of dexamethasone in a small prospective series of 9 patients treated with the DCyBorSeq regimen [13]. NT-proBNP levels doubled after dexamethasone administration suggesting that dexamethasone increases left ventricular pressure by increasing congestion. In the present study we show that early dexamethasone administered combined or alone is associated with important clinical events, related to cardiac decompensation, including but not limited to sudden deaths (Fig 3A and 3B). Therefore, the increase in NT-proBNP levels, 24 h after dexamethasone administration suggests that dexamethasone may induce direct cardiotoxicity.

Palladini *et al.* reported that high-dose dexamethasone was beneficial for treating AL patients and that high-dose dexamethasone administered on D1-4, D9-10, and D17-20 every 35 days was particularly toxic during the induction phase [20]. Of the 5 patients treated with this schedule, two died from arrhythmias. They reported that a milder schedule with 40 mg of high-dose dexamethasone administered on D1-4 of every 21-day cycle reduced toxicity. This early toxicity may partly explain why we observed fewer early deaths when dexamethasone was administered later in the first cycle. Also, the previously cited Sayed et al. study [16] reports that the events observed following chemotherapy (CyBorD) administration and before death were mainly atrioventricular blocks, bradycardia, or arrhythmias. Similarly, we observed that most of the sudden deaths occurred within one or two weeks following the first concomitant administration of CyBorD (Fig 3A). In contrast, when these chemotherapies were introduced sequentially (Fig 3B) or when dexamethasone was delayed (Fig 3C), no sudden deaths occurred during the first chemotherapy cycle. These studies suggested an impact of dexamethasone on cardiac electrical cells and sudden death probably potentiated when dexamethasone is administered concomitantly with cyclophosphamide and bortezomib, in patients with severe cardiac infiltration. These treatments may not have the same negative cardiac effect in patient with mild or no cardiac amyloidosis, as in myeloma patients [21, 22].

We assessed the impact of dexamethasone on cardiac function using the Langendorff heart model in healthy rats. This model allows a careful control of the cardiac responses to dexamethasone, without the confounding effects of other organs and under standardized conditions [23]. We found that dexamethasone significantly reduced LV global function with a decrease in LV contractility and relaxation. It is noteworthy that these acute effects were observed in healthy rats. Thus, in cardiac amyloidosis patients we may expect a more delirious effects on the myocardium with poor prognosis and early deaths.

## Conclusion

We provided preliminary evidence that delaying the administration of dexamethasone during the first cycle of treatment for light chain amyloidosis with severe cardiac involvement reduces early deaths.

## Study limitations

Our study was limited by its retrospective design. This study assessed first-line treatment in AL cardiac amyloidosis patients. Therefore, we did not account for potential changes to therapy or the effect of therapy on biomarkers. Furthermore, in our study there was no control group. Selected AL patients do not receive dexamethasone, but at our institute these are patients with more severe disease and not comparable with those treated with dexamethasone. Electrical events were not recorded before sudden deaths. Sudden deaths can result from various causes: not only electrical cell dysfunction but also pulmonary embolisms and strokes. Therefore, the number of deaths observed in our study when treatments were combined is consistent with that reported by Sayed et. al. [16].

## Clinical perspectives

Competencies in Medical Knowledge: Despite a significant difference in early mortality, long-term survival did not differ between regimens (CyBorDComb, DCyBorSeq, and CyBorDSeq). Our results suggest that early deaths, may be related to the administered chemotherapy. Currently, new therapies for treating AL, such as daratumumab, are under development [24, 25]. It is important that these therapies be assessed as first-line treatment for severe CA patients (NCT 04131309).

Translational outlook: Prospective randomized clinical trials are needed to investigate the role of dexamethasone and optimize the dosage schedule for AL cardiac amyloidosis patients. We suggest that dexamethasone be reduced and delayed or replaced by another corticosteroid.

## Supporting information

**S1 Fig. Chemotherapy regimen and survival according to the year of diagnosis and European stage.** (A) Chemotherapy regimens used in treated patients by year of diagnosis. (B) Survival by year of diagnosis. (C) Chemotherapy regimens used in treated patients by European stages. (D) Survival by European stage.
(TIFF)

**S1 Table. Characteristics and description of cumulative dose of dexamethasone administered to patients who died during the first cycle of chemotherapy.**
(PDF)

## Acknowledgments

The authors would like to thank all the members of the Mondor Amyloidosis Network for their continued collaboration and dedication to care and management of amyloidosis. We would also like to thank Trevor Stanbury (Speak the Speech) for medical writing assistance.

## Author Contributions

**Conceptualization:** Silvia Oghina, Damien Vitiello, Ekaterini Kordeli, Floriane Gilles, Thibaud Damy.

**Data curation:** Mélanie Bézard.

**Formal analysis:** Mélanie Bézard, Damien Vitiello, Floriane Gilles, Jason Shourick, Onnik Agbulut, Thibaud Damy.

**Funding acquisition:** Thibaud Damy.

**Investigation:** Mélanie Bézard, Silvia Oghina, Damien Vitiello, Mounira Kharoubi, Arnault Galat, Amira Zaroui, Soulef Guendouz, Floriane Gilles, David Hamon, Vincent Audard, Emmanuel Teiger, Elsa Poullot, Valérie Molinier-Frenkel, François Lemonnier, Onnik Agbulut, Fabien Le Bras, Thibaud Damy.

**Methodology:** Mélanie Bézard, Damien Vitiello, Floriane Gilles, Jason Shourick, Onnik Agbulut, Thibaud Damy.

**Resources:** Mounira Kharoubi, Onnik Agbulut.

**Supervision:** Damien Vitiello, Onnik Agbulut, Thibaud Damy.

**Validation:** Damien Vitiello, Floriane Gilles, Onnik Agbulut, Thibaud Damy.

**Writing – original draft:** Mélanie Bézard, Damien Vitiello, Floriane Gilles, Thibaud Damy.

**Writing – review & editing:** Mélanie Bézard, Silvia Oghina, Damien Vitiello, Mounira Kharoubi, Ekaterini Kordeli, Arnault Galat, Amira Zaroui, Soulef Guendouz, Floriane Gilles, Jason Shourick, David Hamon, Vincent Audard, Emmanuel Teiger, Elsa Poullot, Valérie Molinier-Frenkel, François Lemonnier, Onnik Agbulut, Fabien Le Bras, Thibaud Damy.

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
