## [Decision Letter · Decision Letter 0]

27 May 2021

PONE-D-21-15097

Dexamethasone is associated with early deaths in light chain amyloidosis patients with severe cardiac involvement

PLOS ONE

Dear Dr. Bézard,

Thank you for submitting your manuscript to PLOS ONE. After careful consideration, we feel that it has merit but does not fully meet PLOS ONE’s publication criteria as it currently stands. Therefore, we invite you to submit a revised version of the manuscript that addresses the points raised during the review process.

We look forward to receiving your revised manuscript.

Kind regards,

Corstiaan den Uil

Academic Editor

PLOS ONE

Journal Requirements:

[Pr Vincent Audard received consulting fees from Addmedica not related to the submitted work. Dr Silvia Oghina reported personal fees from Pfizer, outside of the submitted work. Pr Thibaud Damy received grant and/or consulting fees from PFIZER, AKCEA, ALNYLAM, PROTHENA, and JANSSEN outside the submitted work.

The other authors declared no conflict of interests.].

Reviewers' comments:

Reviewer's Responses to Questions

**Comments to the Author**

1. Is the manuscript technically sound, and do the data support the conclusions?

Reviewer #1: Partly

2. Has the statistical analysis been performed appropriately and rigorously? 

Reviewer #1: Yes

3. Have the authors made all data underlying the findings in their manuscript fully available?

Reviewer #1: No

4. Is the manuscript presented in an intelligible fashion and written in standard English?

Reviewer #1: Yes

5. Review Comments to the Author

Reviewer #1: Bézard et al. describe an association between dexamethasone and early deaths in AL amyloidosis. This is an interesting topic as dosages and schedules of dexamethasone substantially differ between treatment schemes in AL amyloidosis and secondly the impact of cardiac toxicity/mortality of dexamethasone in AL amyloidosis is unclear. Even with novel therapies dexamethasone dosages and schedules vary between treatment recommendations for AL. The manuscript in the current version cannot convince me to delay dexamethasone during chemotherapy. Nevertheless, with the three different CyBorD regimens and the multivariate analysis focusing on time of dexamethasone application the authors can show a direct correlation between dexamethasone application and fatalities. This does not seem to be related to bortezomib or low dose cyclophoshamide. It seems to me that first dexamethasone application is a crucial, but also dangerous event for cardiac AL patients. We might harm some patients with dexa, but others will benefit in the long run. The authors should focus on this result of their study and highlight it within the conclusion of the abstract. It would be beneficial to further describe the early deaths associated with dexamethasone (dosages and stages).

These are my additional comments regarding the manuscript:

Abstract:

• Line 44/45: I was at first struggling with the definition of the primary endpoint: cardiovascular mortality and cardiac transplantation at days 22 and 455. Why were day 22 and day 455 primarily chosen? Month 3 as mentioned within the background of the abstract is from a clinical perspective a better option to compare early deaths. Furthermore, merging cardiovascular mortality and cardiac transplantation seems problematic at day 455. For some patients with severe cardiac AL, transplantation might be a success as transplant shortages cause waiting list periods of several months.

Manuscript:

• Line 144/145: “dexamethasone orally at a dose of either 10, 20, or 40mg, adapted to cardiac severity.” Since this is a retrospective analysis assessing the impact of dexamethasone on cardiotoxicity, the standard dosage should be mentioned. Furthermore, the adaption mechanism should be described.

• Line 205-207 What lead to this significant difference between treatment groups?

• Line 207-209: The group of living patients at day 455 had significantly lower cardiac biomarkers. I would like to see a comparison between living/deceased MayoIIIa and Mayo IIIb patients regarding dexamethasone dosages at day 22 and day 455

• Line 258: Death associated with younger age is probably related to a center effect. Please shortly mention the center effect within your manuscript.

• Line 388-390 + Figure 1: There were 18 patients treated at your institution without dexamethasone and 22 patients treated with other chemotherapy than CyBorD. Did these patients all receive different steroids like prednisone? Did any of these patients only receive cyclophosphamide or cyclophosphamide+ bortezomib without dexamethasone? If yes, did any of them die before day 15? These cases should be included within CyBorDSeq. If no state this information within your manuscript as it will support the message that dexamethasone induces some cardiac deaths.

• Figure 3: What dosages of dexamethasone were given to the patients dying between day 1 and day 22 in A and B and on day 29 in C? Was it only stage IIIb patients?

6. PLOS authors have the option to publish the peer review history of their article (what does this mean?). If published, this will include your full peer review and any attached files.

Reviewer #1: **Yes: **Christoph R. Kimmich

---

## [Author Response · Author response to Decision Letter 0]

11 Aug 2021

Dear Doctor Chenette, dear reviewer,

We thank you for your interest in our work and for the time you have devoted to us.

Following your advice, we have listed each of the Editor/Reviewer comments with our response being added underneath each one. We have added an exact copy of the revised text from the manuscript (where appropriate) beneath each response. 

Further, we completed the Competing Interests section, as followed: Pr Vincent Audard received consulting fees from Addmedica not related to the submitted work. Dr Silvia Oghina reported personal fees from Pfizer, outside of the submitted work. Pr Thibaud Damy received grant and/or consulting fees from PFIZER, AKCEA, ALNYLAM, PROTHENA, and JANSSEN outside the submitted work. The other authors declared no conflict of interests. 

Hoping that these answers give you complete satisfaction,

Yours sincerely

Mélanie Bézard 

Corresponding author

---

## [Editor Report · Decision Letter 1]

26 Aug 2021

Dexamethasone is associated with early deaths in light chain amyloidosis patients with severe cardiac involvement

PONE-D-21-15097R1

Dear Dr. Bézard,

We’re pleased to inform you that your manuscript has been judged scientifically suitable for publication and will be formally accepted for publication once it meets all outstanding technical requirements.

Kind regards,

Corstiaan den Uil

Academic Editor

PLOS ONE
---

## [Editor Report · Acceptance letter]

6 Sep 2021

PONE-D-21-15097R1 

Dexamethasone is associated with early deaths in light chain amyloidosis patients with severe cardiac involvement 

Dear Dr. Bézard:

I'm pleased to inform you that your manuscript has been deemed suitable for publication in PLOS ONE. Congratulations! Your manuscript is now with our production department. 

Kind regards, 

on behalf of

Dr. Corstiaan den Uil 

Academic Editor

PLOS ONE